# Deflagration-to-Detonation Transition in Stochiometric Propane–Hydrogen–Air Mixtures

**Igor O. Shamshin** [1] , **Maxim V. Kazachenko** [1], **Sergey M. Frolov** [1,2,*] and **Valentin Y. Basevich** [1]

1 Department of Combustion and Explosion, Semenov Federal Research Center for Chemical Physics of the Russian Academy of Sciences, 4 Kosygin Str., Moscow 119991, Russia
2 Merzhanov Institute of Structural Macrokinetics and Materials Science of the Russian Academy of Sciences, 8 Acad. Osipyan Str., Chernogolovka 142432, Russia
* Correspondence: smfrol@chph.ras.ru

**Abstract:** Hydrocarbon–hydrogen blends are often considered as perspective environmentally friendly fuels for power plants, piston engines, heating appliances, home stoves, etc. However, the addition of hydrogen to a hydrocarbon fuel poses a potential risk of accidental explosion due to the high reactivity of hydrogen. In this manuscript, the detonability of stoichiometric $C_3H_8$–$H_2$–air mixtures is studied experimentally in terms of the run-up time and distance of deflagration to detonation transition (DDT). The hydrogen volume fraction in the mixtures varied from 0 to 1. Three different configurations of detonation tubes were used to ensure the DDT in the mixtures of the various compositions. The measured dependences of the DDT run-up time and distance on the hydrogen volume fraction were found to be nonlinear and, in some cases, nonmonotonic with local maxima. Blended fuel detonability is shown to increase sharply only at a relatively large hydrogen volume fraction (above 70%), i.e., the addition of hydrogen to propane in amounts less than 70% vol. does not affect the detonability of the blended fuel significantly. The observed nonlinear/nonmonotonic dependences are shown to be the manifestation of the physicochemical properties of hydrogen-containing mixtures. An increase in the hydrogen volume fraction is accompanied by effects leading to both an increase and a decrease in mixture sensitivity to the DDT. Thus, on the one hand, the increase in the hydrogen volume fraction increases the mixture sensitivity to DDT due to an increase in the laminar flame velocity and a decrease in the self-ignition delay at isotherms above 1000 K and pressures relevant to DDT. On the other hand, the mixture sensitivity to DDT decreases due to the increase in the speed of sound in the hydrogen-containing mixture, thus leading to a decrease in the Mach number of the lead shock wave propagating ahead of the flame, and to a corresponding increase in the self-ignition delay. Moreover, for $C_3H_8$–$H_2$–air mixtures at isotherms below 1000 K and pressures relevant to DDT, the self-ignition delay increases with hydrogen volume fraction.

**Keywords:** $C_3H_8$–$H_2$–air mixtures; deflagration-to-detonation transition; run-up distance; run-up time





## 1. Introduction

This work continues the research started in [1–4], where experiments on deflagration-to-detonation transition (DDT) in stoichiometric $CH_4$–$H_2$–air [1,2] and $C_2H_4$–$H_2$–air [3,4] mixtures with a hydrogen volume fraction $x_{H2}$ ranging from 0 to 1 were conducted in tubes with three different configurations at normal pressure and temperature (NPT) conditions. The DDT run-up time, $\tau_{DDT}$, and distance, $L_{DDT}$, for $CH_4$–$H_2$–air mixtures were shown to change nonmonotonically with $x_{H2}$: $\tau_{DDT}(x_{H2})$ and $L_{DDT}(x_{H2})$ dependences showed local maxima in the interval $0.25 < x_{H2} < 0.65$, i.e., the addition of hydrogen to the $CH_4$–air mixture could worsen its detonability contrary to expectations. As for $C_2H_4$–$H_2$–air mixtures, their detonability was shown to increase sharply only at a relatively large hydrogen content (at $x_{H2} > 0.7$).

This study deals with stoichiometric $C_3H_8$–$H_2$–air mixtures with a hydrogen volume fraction $x_{H2}$, also ranging from 0 to 1. Such mixtures are used in power plants, piston

engines, heating appliances, home stoves, etc., operating on liquefied petroleum gas, which mainly consists of propane and butane.

Laminar flames and self-ignition of $C_3H_8$–air and $H_2$–air mixtures were studied by many researchers. The laminar flame speeds in $C_3H_8$–air and $H_2$–air mixtures were measured in [5–9] and [10–12], respectively. The self-ignition delays of undiluted $C_3H_8$–air mixtures were measured in [13] behind reflected shock waves at initial pressure $P_0 = 2$ and 20 atm, initial temperature $T_0 = 1000$–1750 K, and equivalence ratios $\Phi = 0.5$, 1.0, and 2.0. Self-ignition delays were also measured in [14] in a rapid compression machine at $P_0 = 21$, 27, and 37 atm, $T_0 = 680$–970 K, and $\Phi = 0.5$, 1.0, and 2.0. For $H_2$–air mixtures, self-ignition delays behind reflected shock waves and in rapid compression machines were measured in [15,16] at $P_0 = 1$–70 atm; $T_0 = 914$–2200 K, and $\Phi = 0.1$–4.0.

In $C_3H_8$–$H_2$–air mixtures, the laminar flame speed was measured in a spherical bomb [17] at $\Phi = 1.0$, $x_{H2} = 0.063$ and 0.154; $P_0$ and $T_0$ were varied from 1 to 7 atm and from 300 to 500 K, respectively. It was shown that the dependence of the laminar flame speed on pressure along the isentrope of the fresh mixture had two maxima. After ignition, the flame accelerated and reached the first maximum value at a pressure of about 1.5–2 atm. Thereafter, the flame slowed down and reached a minimum at a pressure of about 2.5–3 atm and accelerated again, reaching the second maximum value at a pressure of about 4–6 atm. In [18], the laminar flame speeds in $C_3H_8$–$H_2$–air mixtures with small hydrogen content were measured at $\Phi = 0.6$–1.9; $P_0 = 0.1$ MPa; and $T_0 = 298$ K. The laminar flame speed was shown to increase linearly with $x_{H2}$. Inhibition of combustion of $H_2$–air mixtures by small additives of $C_3H_8$ under NPT conditions was studied experimentally and theoretically in [19]. The addition of 15% vol. $C_3H_8$ to the $H_2$–air mixture with $\Phi = 1.0$ led to a decrease in the laminar flame speed by approximately a factor of 3: from 2 to 0.7 m/s. The speeds of laminar spherically diverging flames in $C_3H_8$–$H_2$–air mixtures were measured in [20] at $P_0 = 0.1$ MPa, $T_0 = 298$ K, $\Phi = 0.6$–1.6, and $x_{H2} = 0$–1. The addition of $H_2$ was shown to exert a significant effect on the laminar flame speed only at large $H_2$ volume fractions in the mixture. Thus, when 80% vol. $H_2$ was added, the flame speed in a stoichiometric mixture increased by a factor of about 2 in comparison with a stoichiometric $C_3H_8$–air mixture without $H_2$ addition: from 0.4 to 0.7 m/s.

The authors of [21] studied experimentally and numerically the laminar flame speeds of fuel lean $C_3H_8$–$H_2$–air mixtures at $P_0 = 0.1$ MPa, $T_0 = 298$ K, $\Phi = 0.45$–0.65, and $x_{H2} = 0.95$ in the counterflow configuration. The results of calculations were shown to correlate well with measurements. Combustion of stoichiometric homogeneous liquified petroleum gas—air mixtures with $0 \leq x_{H2} \leq 0.5$ was studied experimentally in [22]. An extended kinetic model, including both high- and low-temperature mechanisms of $C_3H_8$ oxidation was used in [23] to perform kinetic calculations of combustion and self-ignition of $C_3H_8$–$H_2$–air mixtures. Addition of $H_2$ to $C_3H_8$ was shown to increase the laminar flame speed and expand the concentration flammability limits.

The data on measurements and calculations of the self-ignition delays of $C_3H_8$–$H_2$–air mixtures were reported in [23–25]. Calculations in [23] showed that, at relatively low initial temperatures, the self-ignition delay time of $C_3H_8$–$H_2$–air mixtures could be longer than that of a pure $C_3H_8$–air mixture; conversely, at relatively high temperatures, small additives of $C_3H_8$ to the $H_2$–air mixture accelerated self-ignition. In [24,25], the results of measurements and calculations of self-ignition delays for $C_3H_8$–$H_2$–air mixtures were reported for $P_0 = 1.2$, 4.0, and 10 atm, $T_0 = 1000$–1600 K, $\Phi = 0.3$–1, $x_{H2} = 0$–1, and volume fraction of diluent gas $\alpha = 0.9132$–0.943 [24] and for $P_0 = 5.2$–11.2 bar, $T_0 = 920$–1900 K, $\Phi = 1.0$, and $x_{H2} = 0$, 0.3, 0.5, and 0.7 [25]. The addition of $H_2$ had a significant effect on the self-ignition delay only at a large $H_2$ volume fraction in the mixture: the addition of 70% $H_2$ decreased the self-ignition delay twofold. It was shown in [26], based on kinetic calculations for $T_0 = 1000$–2000 K and $P_0 = 0.05$–200 atm, that the self-ignition delay of a stoichiometric $C_3H_8$–$H_2$–air mixture decreased with $x_{H2}$, although at high pressures and low temperatures, the addition of a large amount of $H_2$ had almost no effect on the self-ignition delay.

The developed detonations in $C_3H_8$–$H_2$–air mixtures with $\Phi$ = 0.8–2.2 and $x_{H2}$ = 0.4, 0.5, 0.7, and 1.0 at NPT conditions were studied in [27], both experimentally and computationally, in a smooth-walled tube 10 mm in diameter and 6 m long, with one open end. An electric spark and a Shchelkin spiral were used to ignite the mixture and to ensure the DDT, respectively. It was shown that the Chapman–Jouguet (CJ) one-dimensional theory was applicable to the detonation of such blended fuels, as the measured detonation velocities were consistent with the calculated values. The cell sizes of developed detonations in stoichiometric $C_3H_8$–$H_2$–air mixtures with $0 \leq x_{H2} \leq 1.0$ at NPT conditions were measured in [28] by the smoked foil technique, using a detonation tube 100 mm in diameter and 2.035 m in length. Detonation was initiated by transmitting a detonation wave from the donor tube 42 mm in diameter and 788 mm in length filled with the stoichiometric $C_3H_8$–$O_2$ mixture. The measured cell sizes were shown to vary from 75 to 25 mm when $x_{H2}$ was varied from 0 to 1, namely from pure $C_3H_8$ to pure $H_2$. Importantly, the cell size was almost constant (~35–40 mm) at $0.2 \leq x_{H2} \leq 0.8$. Detonation cell sizes in $C_3H_8$–$H_2$–air mixtures with $\Phi$ = 0.7–2.0 and $0.5 \leq x_{H2} \leq 1$ at NPT conditions were also measured by the smoked foil technique in [29], using tubes 52 and 92 mm in diameter and 9–12 m in length. The measured average detonation cell size for the stoichiometric mixture was shown to vary from 40 mm at $x_{H2}$ = 0.7 to 20 mm at $x_{H2}$ = 0.95.

The literature also contains studies of deflagrations in $C_3H_8$–$H_2$–air mixtures, which produced overpressures lower than detonations. Experiments on suppression of spark-ignited deflagrations in stoichiometric $C_3H_8$–$H_2$–air mixtures with $x_{H2}$ = 0, 0.3, 0.6, 0.9, and 1.0 at NPT conditions were conducted in [30] in a closed tube 70 mm in diameter and uniformly filled with dense rolls of mesh aluminum alloy. It was shown that admixing of $C_3H_8$ to $H_2$ could effectively reduce the overpressure and velocity of the deflagration-induced shock wave. However, at $x_{H2}$ >0.72, the explosion pressure was increased significantly so that the method under study failed to suppress explosion.

The objective of this work is to study DDT in stoichiometric $C_3H_8$–$H_2$–air mixtures with $x_{H2}$ ranging in the entire interval from 0 to 1 applying the method of [31,32]. The specific features of the DDT in such mixtures were not studied in full detail. The effect of the scale factor on the DDT limits in $C_3H_8$–$H_2$–air mixtures in tubes 151 and 54 mm in diameter with a set of annular orifices was studied experimentally in [33]. The study was limited by the mixtures of stoichiometric composition with $x_{H2}$ > 0.5. Experimental investigations of DDT in $C_3H_8$–$H_2$–air mixtures with $\Phi$ = 0.8–1.8 and $0.5 \leq x_{H2} \leq 1$ at NPT conditions were also performed in [34,35] using tubes 52 and 92 mm in diameter and 9–12 m in length. A Shchelkin spiral was used to promote flame acceleration and detonation onset. At $\Phi$ = 1, the DDT run-up distance, $L_{DDT}$, was shown to decrease with $x_{H2}$ from 1.3 m at $x_{H2}$ = 0.7 to 0.7 m at $x_{H2}$ = 0.95 [35]. Importantly, the dependence $L_{DDT}(x_{H2})$ was highly nonlinear: the slope of the curve increased with $x_{H2}$. In $C_3H_8$–air mixtures, DDT at NPT conditions was obtained in tubes of relatively large size with regular obstacles (see, e.g., [36]). In $H_2$–air mixtures, DDT at NPT conditions was obtained in tubes of relatively small size (see, e.g., [37]). The aim and the results of this work are the novel and distinctive features of the present paper.

## 2. Materials and Methods

### 2.1. Test Facility

The test facility was previously described in detail in [2,4,31,32]. Briefly, the facility was a 50 mm diameter detonation tube with one open end. It consisted of a prechamber, flame acceleration section, helical section, and measurement section. It was equipped with ignition, control, and data acquisition systems. A $C_3H_8$–$H_2$–air mixture was prepared by partial pressures in a forty-liter mixer according to the reaction equation:

$$(1 - x_{H2})C_3H_8 + x_{H2}H_2 + (5 - 4.5x_{H2})(O_2 + 3.762N_2) = (4 - 3x_{H2})H_2O + 3(1 - x_{H2})CO_2 + 3.762(5 - 4.5x_{H2})N_2, \quad (1)$$

and filled the tube through the prechamber equipped with a spark plug. The flame acceleration section included a Shchelkin spiral. The spiral was 940 mm long. It was

made of a steel wire 6.7 mm in diameter with a pitch of 24 mm. The helical section provided gas-dynamic focusing of pressure waves generated by the accelerating flame. It appeared as a double-coil tube. The outer and inner diameters of the coil were 136 and 36 mm, and the coil pitch was 220 mm. The measurement section was a straight tube with smooth walls equipped with pressure sensors (PSs) and ionization probes (IPs) in multiple measurement ports.

Similar to [2,4], experiments with $C_3H_8$–$H_2$–air mixtures were conducted in tubes of several configurations, shown in Figure 1. Configuration C2 differed from C1 by the length of the section between the Shchelkin spiral and the helical section (it was increased by 240 mm). Configuration C3 differed from C1 in that the Shchelkin spiral was shifted from the prechamber to the helical section.

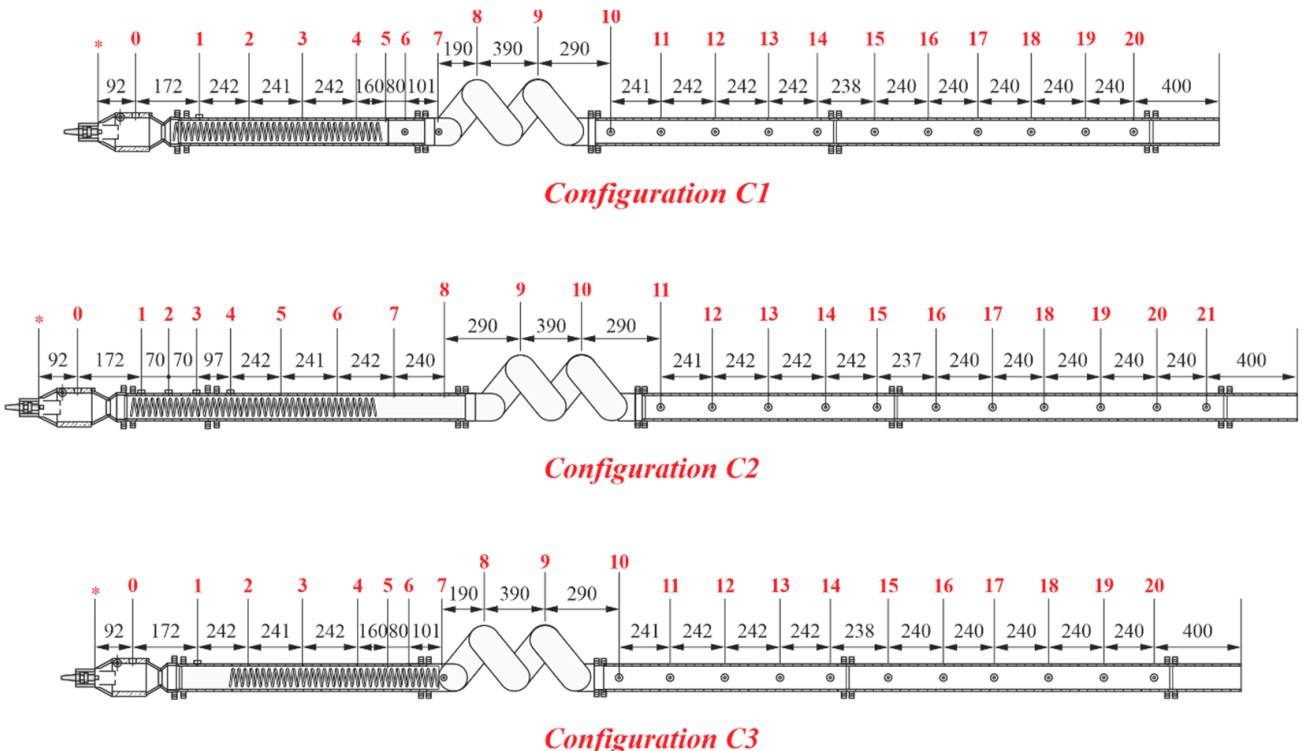

**Figure 1.** Diagrams of detonation tube configurations with measurement segments: * denotes a spark location.

*2.2. Experimental Procedures*

The tube was first blown through with the mixture volume four times the tube volume. The filling process was controlled by the pressure drop in the mixer and by the inlet valve. Ignition was triggered 4–5 s after the inlet valve was closed. The number of successive "shots" with a mixture of fixed composition was usually equal to five. The pressure in the detonation waves was measured by PCB113V24 sensors (natural frequency 500 kHz). The error in measuring the shock wave positions was estimated at ±6 mm. The motion of the leading point of a reaction front was detected using IPs. The error in the leading point position was estimated at ±2 mm. The apparent velocities of the lead pressure wave, $D = D_s$, and the leading point of the reaction front, $D = D_f$, at a measurement segment were determined based on the known distance between the corresponding measurement ports and the time intervals between their arrival at the ports. The error in the apparent velocities at $D > 1000$ m/s was estimated at 3%.

Measured positions and arrival times of pressure waves and reaction fronts allowed the plotting of "time–distance" ($t$–$x$) and "wave velocity—distance" ($D$–$x$) diagrams of the process development, and the DDT run-up time, $\tau_{DDT}$, and distance, $L_{DDT}$ to be obtained. The values of $\tau_{DDT}$ and $L_{DDT}$ were found as the time from ignition and the distance from

the igniter, when the lead shock wave and the reaction front matched within $\pm 6$ µs and their velocities $D_s$ and $D_f$ attained values close to the CJ detonation velocity $D_{CJ}$ (within $\pm 3\%$) for the tested mixture, and the coupled wave propagated steadily at such a velocity in the measurement section. Errors in $\tau_{DDT}$ and $L_{DDT}$ were estimated in terms of the short-to-short scatter. Extrapolation of the retonation wave trajectory was also used to refine the $L_{DDT}$ and $\tau_{DDT}$ values.

## 3. Results

Figures 2–4 provide the $D-x$ diagrams of the DDT process development in $C_3H_8$–$H_2$–air mixtures with $x_{H2}$ = 0, 0.2, 0.4, 0.6, 0.8, and 1.0 in a tube with three different configurations: C1, C2, and C3. Different shots are shown by different symbols. Empty symbols correspond to the velocity of the reaction-front leading point, $D_f$, whereas filled symbols correspond to the velocity of the pressure wave, $D_s$. The vertical dash-and-dot lines mark the ends of the helical section, while the vertical dashed lines mark the ends of the Shchelkin spiral. In some cases, in addition to the line $D_{CJ}$, the second line $D_{s1}$ is plotted, which corresponds to the measured velocity of nonideal detonation inside the Shchelkin spiral. The gray vertical bar corresponds to the measured DDT run-up distance $L_{DDT}$ with the bar width indicating the shot-to-shot scatter in the $L_{DDT}$ value. In addition to Figure 3, plotted for the tube with configuration C2, Figure 5 provides the $D-x$ diagrams of the development of the DDT process in fuel–air mixtures with a small content of $C_3H_8$ (from 5 to 1% vol.), that is with $x_{H2}$ = 0.95, 0.96, 0.97, 0.98, and 0.99 in the tube with configuration C2.

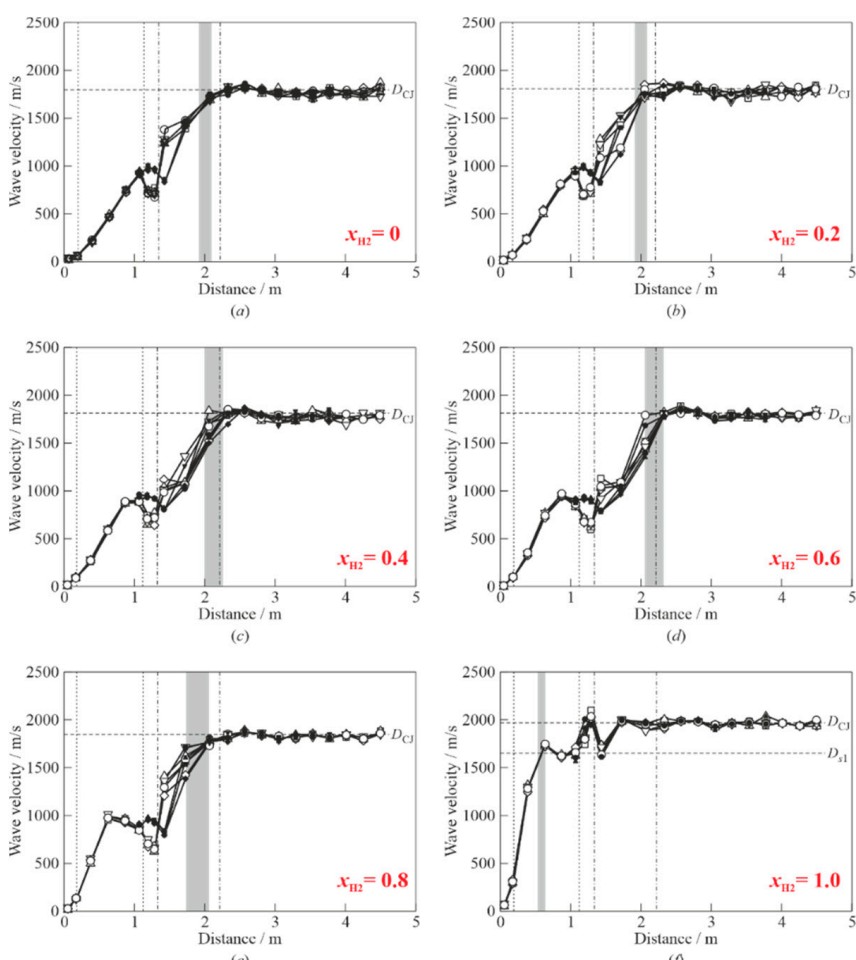

**Figure 2.** $D-x$ diagrams of deflagration-to-detonation transition in a tube with C1 configuration in five shots in mixtures with $x_{H2}$ equal to 0 (**a**), 0.2 (**b**), 0.4 (**c**), 0.6 (**d**), 0.8 (**e**) and 1.0 (**f**).

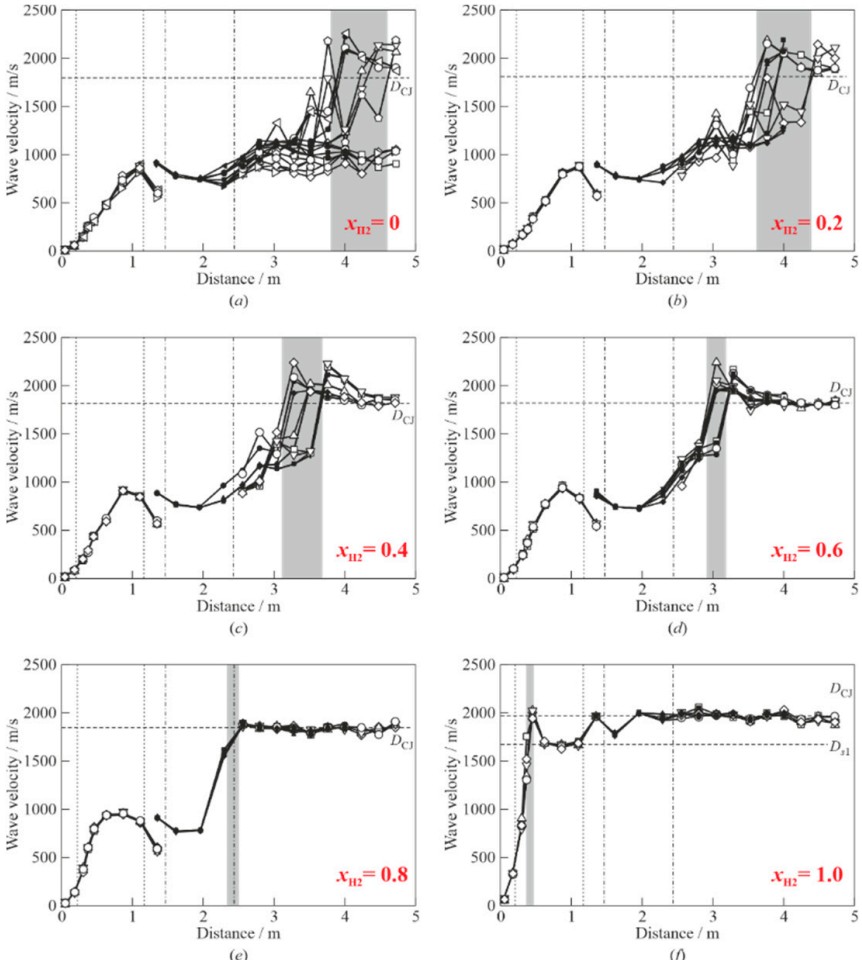

**Figure 3.** *D–x* diagrams of deflagration-to-detonation transition in a tube with C2 configuration in five shots in mixtures with $x_{H2}$ equal to 0 (**a**), 0.2 (**b**), 0.4 (**c**), 0.6 (**d**), 0.8 (**e**) and 1.0 (**f**).

Considering Figures 2–5, the following features can be highlighted:

(1) DDT was recorded in the entire interval $0 \leq x_{H2} \leq 1$ in the pulse-detonation tube for all three configurations: C1, C2, and C3.

(2) For each $x_{H2}$ value, the shot-to-shot dynamics of flame acceleration was well reproduced.

(3) In the tube with configurations C1 and C3, for all values of $x_{H2}$, DDT was registered in the helical section, while in the tube with configuration C2, this only happened at $x_{H2} \geq 0.8$ (at $x_{H2} < 0.8$, DDT occurred in the measurement section).

(4) At $x_{H2} = 1.0$, DDT was first registered in the flame acceleration section, where a quasi-steady nonideal detonation with a large velocity deficit was detected, and then this nonideal detonation transformed to a normal detonation wave in the helical section.

(5) At each $x_{H2}$ value, the detonation velocity in the measurement section was well reproduced in each shot.

(6) Small additives of $C_3H_8$ (from 2 to 5% vol.) to hydrogen–air mixtures inhibited flame acceleration in the section with the Shchelkin spiral: flame acceleration in this section slowed down with the addition of $C_3H_8$, and the maximum acceleration was shifted towards larger distances from the ignition source.

(7) At $C_3H_8$ volume fraction on the level of 1%, the maximum flame acceleration in the section with the Shchelkin spiral was larger than in the stoichiometric hydrogen–air mixture; however, the maximum acceleration was attained at a larger distance from the ignition source (Figure 6). This means that such a small additive of $C_3H_8$ to $H_2$–air mixture promotes turbulent flame acceleration.

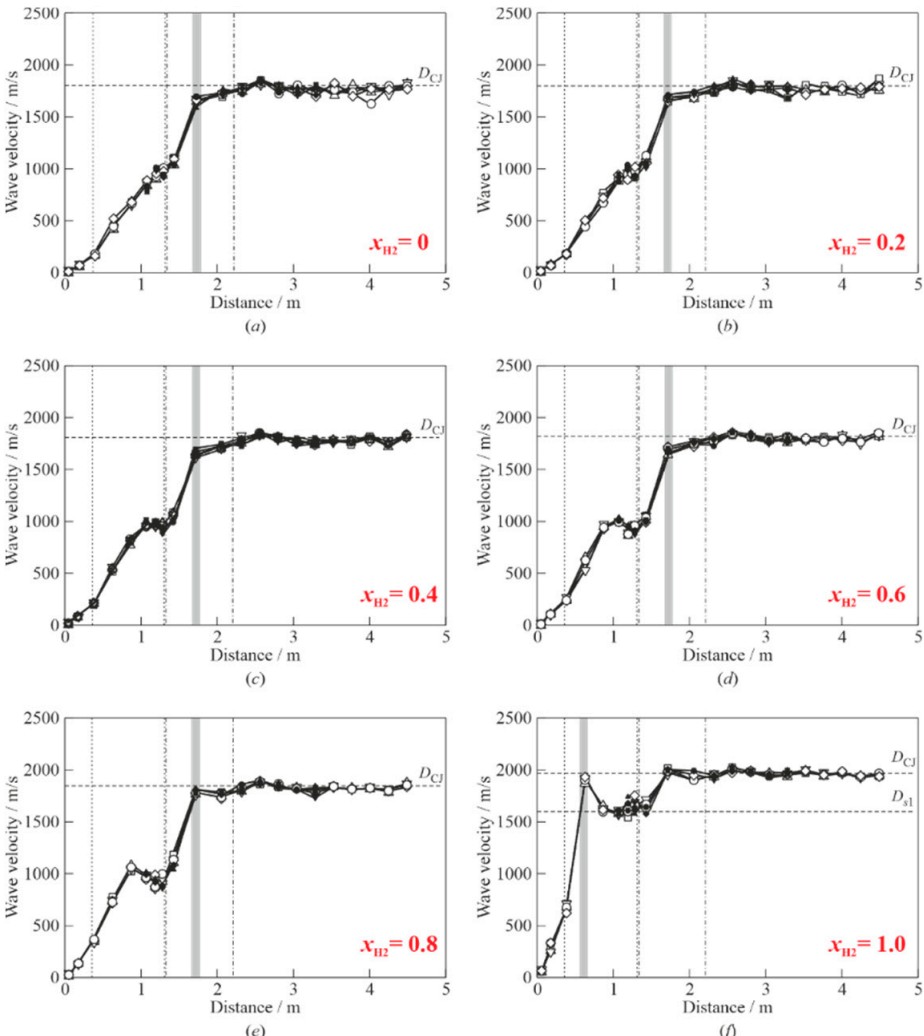

**Figure 4.** *D–x* diagrams of deflagration-to-detonation transition in a tube with C3 configuration in five shots in mixtures with $x_{H2}$ equal to 0 (**a**), 0.2 (**b**), 0.4 (**c**), 0.6 (**d**), 0.8 (**e**) and 1.0 (**f**).

As a result, the overdriven detonation arising during DDT appears to be faster (see arrow in Figure 5e) than in the stoichiometric hydrogen–air mixture (compare Figures 3f and 5e). As was shown theoretically in [23,38], small additives of alkane hydrocarbons could promote self-ignition of $H_2$–air mixtures in a certain temperature range.

Figure 7 provides an example of the *t–x* diagram for the DDT in a $C_3H_8$–$H_2$–air mixture with $x_{H2}$ = 0.4. The gray horizontal bar is the measured value of $\tau_{DDT}$ with a shot-to-shot scatter: 10.3–10.7 ms. Following [39], $\tau_{DDT}$ can be interpreted as the self-ignition delay time of a gas particle drawn into motion and compressed by pressure waves generated by the accelerating flame.

Figure 8 provides an example of primary records of IPs and PSs in a shot with $C_3H_8$–$H_2$–air mixture ($x_{H2}$ = 0.2) in the tube with configuration C1. The intensity of the lead shock wave in measurement ports 4 to 7 located in the flame acceleration section is relatively low, and the time lag between the lead shock wave and the reaction front, $\tau_l$, reaches its maximum value in port 7: $\tau_{l7} \approx 80$ µs (see the records of IP and PS in port 7, and the arrow). The onset of detonation occurs nearby port 8 inside the helical section (see the arrow with a positive slope in Figure 8). The records of PSs in ports 4 to 8 clearly show a retonation wave (see the arrow with a negative slope), which also appears nearby port 8.

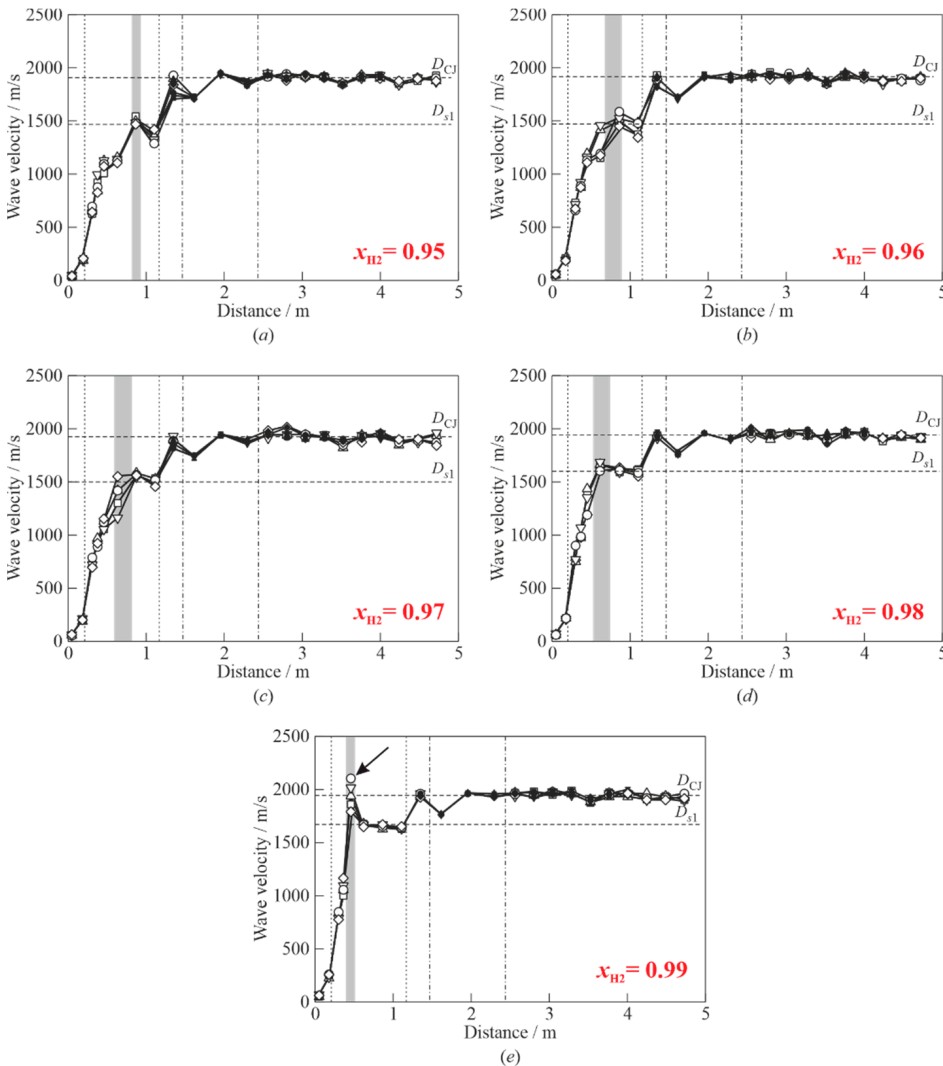

**Figure 5.** *D–x* diagrams of deflagration-to-detonation transition in a tube with C2 configuration in five shots in mixtures with $x_{H2}$ equal to 0.95 (**a**), 0.96 (**b**), 0.97 (**c**), 0.98 (**d**) and 0.99 (**e**).

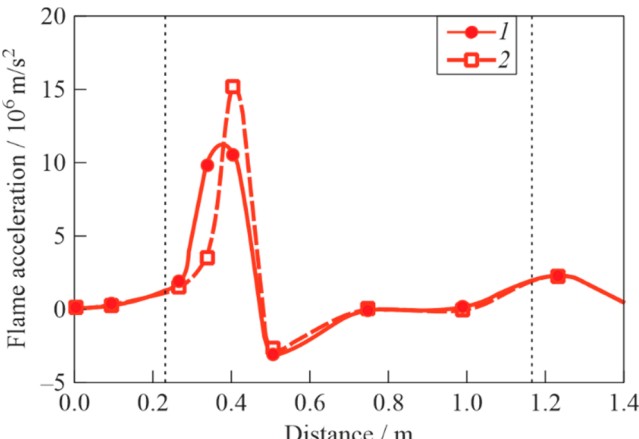

**Figure 6.** Flame acceleration vs. distance during DDT in mixtures with $x_{H2}$ = 1.0 (1) and 0.99 (2) in a tube with C2 configuration (averaged over five shots).

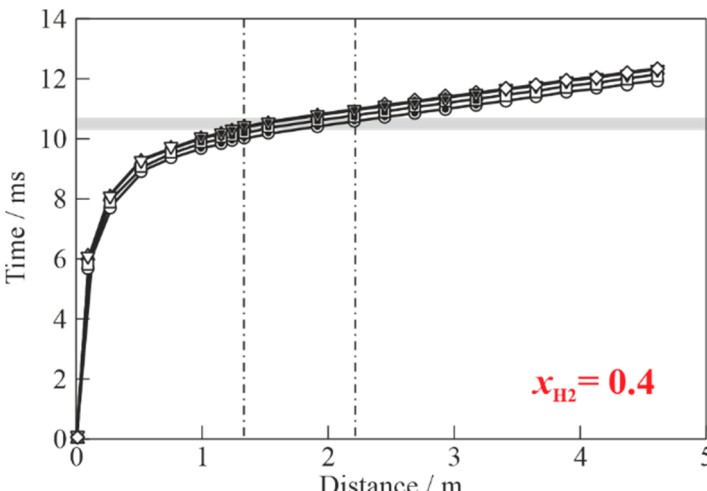

**Figure 7.** $t - x$ diagram for the DDT process in a mixture with $x_{H2} = 0.4$ in five successive shots in a tube with C3 configuration.

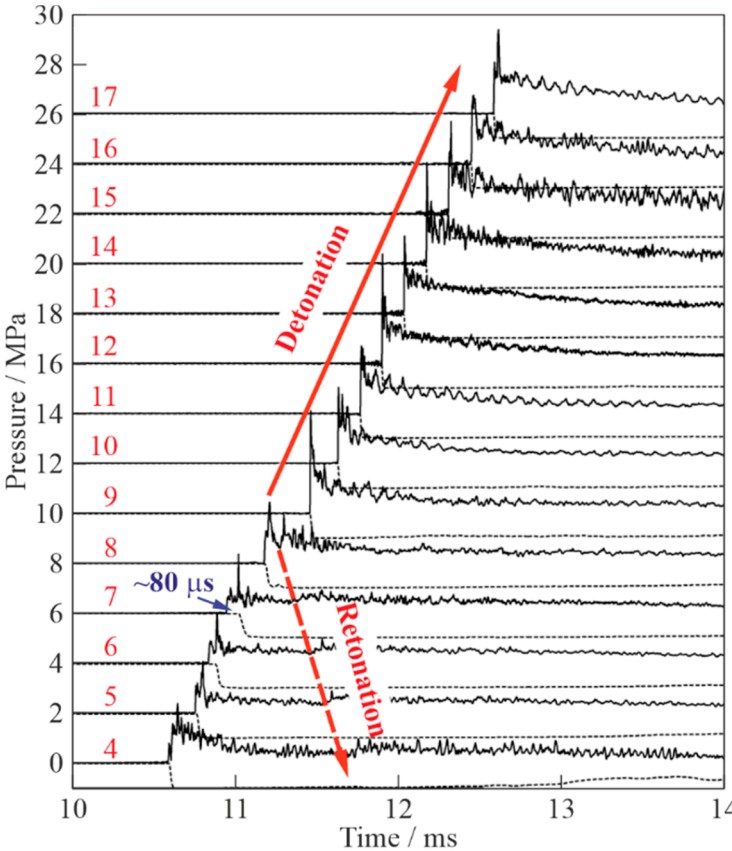

**Figure 8.** Records of IPs (dotted curves) and PSs (solid curves) in measurement ports 4 to 17 in a single shot for a $C_3H_8$–$H_2$–air mixture with $x_{H2} = 0.2$ in a tube with C1 configuration.

## 4. Discussion

The data presented in Figures 2–8 are summarized in Figures 9 and 10. Figure 9 presents the dependences $L_{DDT}(x_{H2})$ (Figure 9a) and $\tau_{DDT}(x_{H2})$ (Figure 9b) obtained in a tube with three configurations: C1, C2, and C3. For the sake of comparison, Figure 10a,b present similar dependences for the stoichiometric $CH_4$–$H_2$–air mixtures in a tube with same configurations [2]. Comparison shows that the DDT in stoichiometric $C_3H_8$–$H_2$–air mixtures exhibits similar features to those of the DDT in stoichiometric methane–hydrogen–

air mixtures: the dependences $L_{DDT}(x_{H2})$ and $\tau_{DDT}(x_{H2})$ are nonlinear, so that mixture detonability in terms of the DDT run-up distance and time begins to increase sharply only at a relatively large hydrogen content (at $x_{H2} > 0.7$). Moreover, in the tube with configuration C1, the dependence $L_{DDT}(x_{H2})$ for $C_3H_8$–$H_2$–air mixtures is nonmonotonic: at $0.20 < x_{H2} < 0.60$, the DDT run-up distance $L_{DDT}$ increases with $x_{H2}$ and reaches a maximum value of $2.2 \pm 0.2$ m at $x_{H2} = 0.6$. In the tube with the same configuration (C1), the dependences $L_{DDT}(x_{H2})$ and $\tau_{DDT}(x_{H2})$ for methane–hydrogen–air mixtures were also nonmonotonic [2]: the maximum values of $L_{DDT}$ ($3.7 \pm 1.2$ m) and $\tau_{DDT}$ ($11.2 \pm 1.2$ ms) were achieved at $x_{H2} = 0.35$. In the tube with configuration C2, the DDT in the methane–hydrogen–air mixture with $x_{H2} < 0.5$ was not detected. The fact that $L_{DDT}$ decreases with $x_{H2}$ at $x_{H2} > 0.7$ monotonically but nonlinearly is consistent with earlier findings in [35].

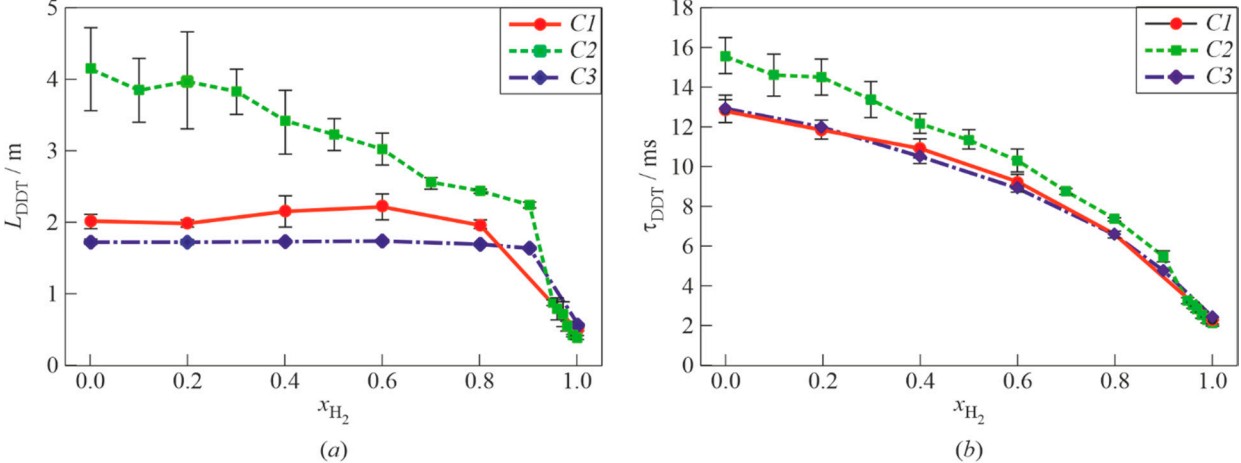

**Figure 9.** Measured $L_{DDT}$ (**a**) and $\tau_{DDT}$ (**b**) vs. $x_{H2}$ in stoichiometric $C_3H_8$–$H_2$–air mixtures: curves correspond to tubes with configurations C1 to C3, respectively.

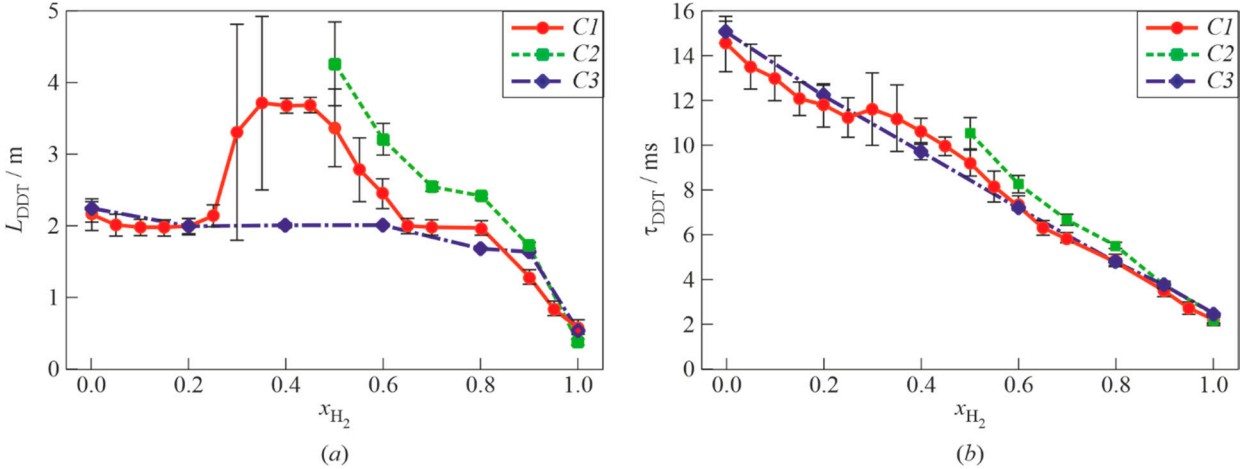

**Figure 10.** Measured $L_{DDT}$ (**a**) and $\tau_{DDT}$ (**b**) vs. $x_{H2}$ in stoichiometric $CH_4$–$H_2$–air mixtures: curves correspond to tubes with configurations C1 to C3, respectively.

Joint consideration of Figures 9 and 10 allows the assertion that the observed effect of nonlinear and nonmonotonic dependences $L_{DDT}(x_{H2})$ and $\tau_{DDT}(x_{H2})$ for both $C_3H_8$–$H_2$–air and $CH_4$–$H_2$–air mixtures is a manifestation of their physicochemical properties, rather than a result of the small diameter and special design of the pulse-detonation tube. When the tube configuration is changed by increasing the space between the end of the Shchelkin spiral and the helical section (transition from C1 to C2 configuration), as well as by moving the Shchelkin spiral to the helical section (transition from C1 to C3 configuration), the $L_{DDT}(x_{H2})$ and $\tau_{DDT}(x_{H2})$ curves are not generally affected for

either mixture: despite the nonmonotonicity degenerates, they remain nonlinear. Like many known critical phenomena in chemical physics (chain/thermal explosion, etc. [40]), nonmonotonic dependences $L_{DDT}(x_{H2})$ and $\tau_{DDT}(x_{H2})$ can manifest themselves only near critical conditions, while they are smoothed out or hidden by other dominant effects away from critical conditions.

To better understand and explain the observed dependences $L_{DDT}(x_{H2})$ and $\tau_{DDT}(x_{H2})$, let us consider Figure 11 plotted for the stoichiometric $C_3H_8$–$H_2$–air mixtures. Here, Figure 11a depicts the experimental dependences of shock wave velocities $D_{s,6-7}(x_{H2})$ and $D_{s,7-8}(x_{H2})$ at measurement segments 6–7 and 7–8, respectively; it also depicts the time lag between the reaction front and the lead shock wave, $\tau_{l7}(x_{H2})$, in the measurement port 7 in the tube with configuration C1. Figure 11b depicts the experimental dependences of shock wave velocities $D_{s,7-8}(x_{H2})$ and $D_{s,8-9}(x_{H2})$ at measurement segments 7–8 and 8–9, as well as the time lag between the reaction front and the shock wave, $\tau_{l8}(x_{H2})$, in measurement port 8 in the tube with configuration C2. Measurement segments 6–7 and 7–8 in the tube with configuration C1, and measurement segments 7–8 and 8–9 in the tube with configuration C2, are located ahead of the helical section and inside the helical section, respectively (see Figure 1). When leaving the Shchelkin spiral, the turbulent flame is known to decelerate sharply due to a decrease in the level of turbulence [41], and flame deceleration, in turn, leads to weakening of the lead shock wave. That is why $D_{s,7-8}$ is always less than $D_{s,6-7}$ in Figure 11a. Near $x_{H2} = 0.6$, the dependences $D_{s,6-7}(x_{H2})$ and $D_{s,7-8}(x_{H2})$ are seen to exhibit minima with a depth of 30–60 m/s, whereas the dependence $\tau_{l7}(x_{H2})$ exhibits a maximum with a height of 20–40 μs. Interestingly, the minimum values of $D_{s,6-7}$ (905 m/s) and $D_{s,7-8}$ (790 m/s), and the maximum value of $\tau_{l7}$ (95 μs), are attained under conditions (at $x_{H2} \approx 0.6$) when $L_{DDT}$ reaches the maximum value (see Figure 9).

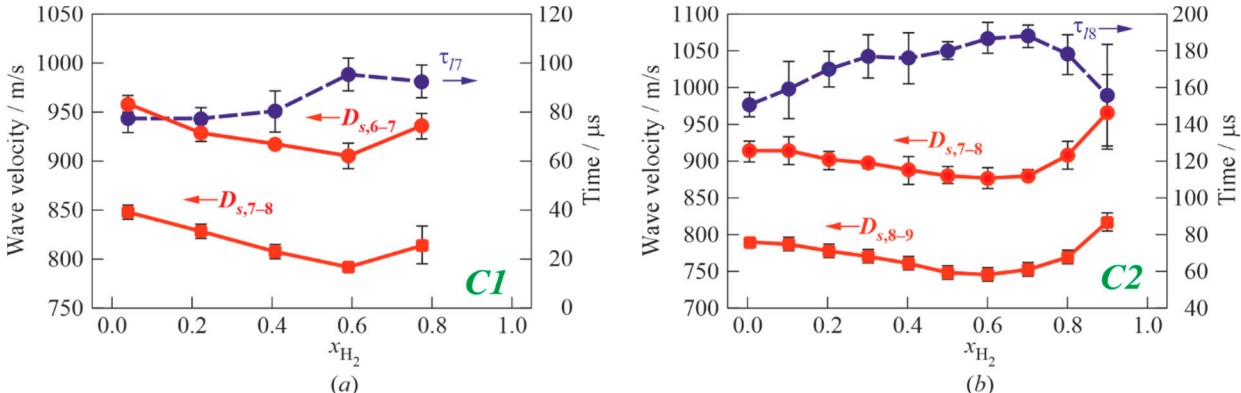

**Figure 11.** Effect of $x_{H2}$ in $C_3H_8$–$H_2$–air mixture on the velocities of shock waves (left scale) and time lags between the shock wave and reaction front (right scale) at measurement segments 6–7 (**a**) and 7–8 (**b**) upstream of the helical section, and 7–8 (**a**) and 8–9 (**b**) inside the helical section in a tube with C1 and C2 configurations.

As for the tube with configuration C2 (see Figure 11b), the corresponding dependences $D_{s,7-8}(x_{H2})$ and $D_{s,8-9}(x_{H2})$ have shallow minima, and the dependence $\tau_{l8}(x_{H2})$ has a shallow maximum, so that the minimum values of $D_{s,7-8}$ (880 m/s) and $D_{s,8-9}$ (730 m/s), and the maximum value of $\tau_{l8}$ (190 μs), are attained at $x_{H2} = 0.4$–0.7.

Recall that in the tube with the C1 configuration, DDT at $x_{H2} = 0.6$ occurs in the helical section, whereas in the tube with the C2 configuration, DDT at $x_{H2} = 0.4$–0.7 occurs further downstream, in the measurement section. The increase in $L_{DDT}$ and $\tau_{DDT}$ in the tube with configuration C2 as compared with that of C1 is apparently associated with a decrease in the velocity of the shock wave in the corresponding measurement segments and, accordingly, with an exponential dependence of the self-ignition delay on the temperature behind a shock wave reflected from the curved wall of the helical section (configuration C1) and behind the lead shock wave traveling in the measurement section (configuration C2).

The question arises of why the curves in Figure 11 behave so unexpectedly in the range $0 < x_{H2} < 0.8$. As mentioned in [1,2], an increase in $x_{H2}$ can be accompanied by effects leading to both an increase and a decrease in mixture sensitivity to DDT. On the one hand, mixture sensitivity must increase as the laminar flame speed $u_n$ increases with $x_{H2}$ [20] and the self-ignition delay $\tau_i$ at isotherms above 1000–1100 K decreases with $x_{H2}$ [23]. In accordance with the theory [42], the flame must accelerate faster and the "explosion in the explosion" [39] must occur earlier. On the other hand, there exist several effects leading to a decrease in mixture sensitivity. Thus, the mixture molecular mass decreases and the speed of sound increases with $x_{H2}$. As a result, the Mach number of the shock wave propagating ahead of the flame in the flame-acceleration section decreases, and so do the temperature and pressure of the shock-compressed gas. Consequently, the self-ignition delay time $\tau_i$ increases. Furthermore, for $C_3H_8$–$H_2$–air mixtures at isotherms below 1000 K, the self-ignition delay increases with $x_{H2}$ [23]. Estimates show that for the minimum values of $D_{s,6-7}$ and $D_{s,7-8}$ in Figure 11a the temperature of the shock-compressed gas behind the lead shock wave is about 750 K, whereas the temperature behind the reflected shock wave is about 930 K, respectively. Following [23], $\tau_i$ increases with $x_{H2}$ under these conditions. An increase in $\tau_i$ with pressure caused by the faster increase in the rate of chain termination compared with the rate of chain branching can also manifest itself [40].

Figure 12 shows the calculated dependences of $\tau_i(x_{H2})$ at temperatures and pressures characteristic of DDT, obtained using a detailed kinetic mechanism [43]. For $C_3H_8$–$H_2$–air mixtures, this effect manifests itself at temperatures above 1200 K. One can see that the isobars 10, 20, and 30 atm in Figure 12 are reversed at 1400 K and $x_{H2} > 0.8$ (marked by the arrow), indicating that the ignition delay increases with pressure, while it decreases with pressure at lower temperatures. It is worth mentioning that the diameter of the tube used herein is close to the limiting tube diameter for $C_3H_8$–air mixtures [37,44]. Transient combustion processes are more sensitive to gas-dynamic disturbances at such conditions. Apparently, the displacement of the DDT location from the helical section to the smooth measurement section in the tube with configuration C2 in the range $0 < x_{H2} < 0.6$ can be caused by a combined effect.

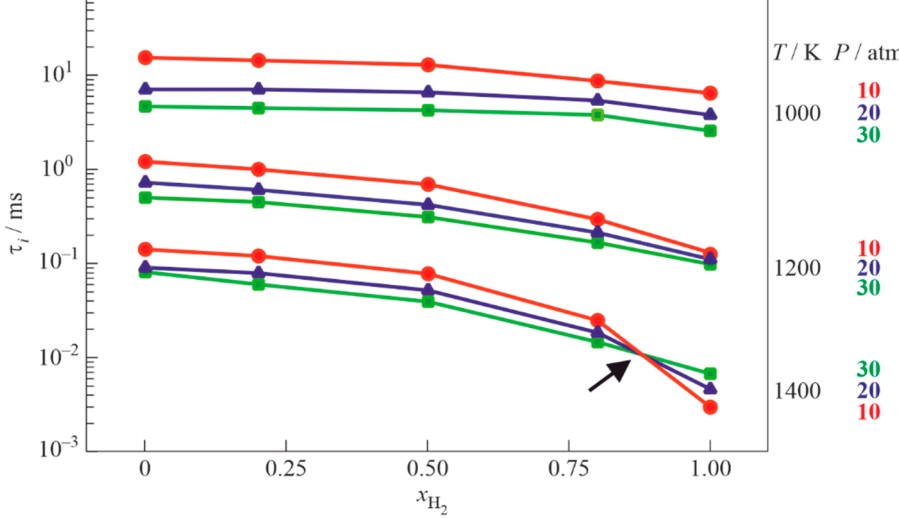

**Figure 12.** Predicted self-ignition delays for stoichiometric $C_3H_8$–$H_2$–air mixtures at different values of $P$, $T$, and $x_{H2}$.

## 5. Conclusions

Propane–hydrogen blends are often considered as perspective environmentally friendly fuels for power plants, piston engines, heating appliances, home stoves, etc. However, admixing of hydrogen to propane poses a potential risk of accidental explosion due to the known high reactivity of hydrogen. In this manuscript, the detonability of stoichiometric

$C_3H_8$–$H_2$–air mixtures is studied experimentally in terms of the DDT run-up time and distance. The hydrogen volume fraction $x_{H2}$ in the mixtures ranged from 0 to 1. The studies were conducted in laboratory-scale detonation tubes with three different configurations. The dependences of the DDT run-up time and distance in such mixtures appeared to be nonlinear and nonmonotonic (in some cases): mixture detonability increased sharply only at relatively large hydrogen content (at $x_{H2} > 0.7$). This means that addition of hydrogen to propane in amounts less than 70% vol. has little effect on the detonability of the blended fuel: despite the DDT run-up time tending to decrease with $x_{H2}$, the DDT run-up distance remains on the level characteristic of the pure propane—air mixture.

The observed unexpected dependences are explained by the physicochemical properties of hydrogen because the tube design modifications did not have much effect on the character of the dependences: although their nonmonotonicity could degenerate, they remained nonlinear. Nonmonotonicity manifests itself only near critical conditions, and it is smoothed out or hidden by other dominant effects away from critical conditions. The observed dependences are explained by the multilateral effects of the addition of hydrogen on the mixture sensitivity to DDT. On the one hand, the increase in the hydrogen volume fraction increases mixture sensitivity to DDT due to the increase in the laminar flame speed and the decrease in the self-ignition delay at isotherms above 1000 K and pressures relevant to DDT. On the other hand, the mixture sensitivity to DDT decreases due to the increase in the speed of sound in the hydrogen-containing mixture, thus leading to a decrease in the Mach number of the lead shock wave propagating ahead of the flame and to a corresponding increase in the self-ignition delay. Moreover, for $C_3H_8$–$H_2$–air mixtures at isotherms below 1000 K and pressures relevant to DDT, the self-ignition delay increases with the hydrogen volume fraction. For the theoretical substantiation of the obtained results, detailed gas-dynamic and kinetic calculations are required.

**Author Contributions:** Conceptualization, S.M.F.; methodology, S.M.F. and I.O.S.; formal analysis, S.M.F. and I.O.S.; investigation, I.O.S., M.V.K. and V.Y.B.; data curation, I.O.S. and M.V.K.; writing—original draft preparation, S.M.F.; writing—review and editing, S.M.F.; supervision, S.M.F.; project administration, S.M.F.; funding acquisition, S.M.F. All authors have read and agreed to the published version of the manuscript.

**Funding:** This research was supported by a subsidy given to Semenov Federal Research Center for Chemical Physics of the Russian Academy of Sciences to implement the state assignment on the topic No. 0082-2019-0006 (Registration No. AAAA-A21-121011990037-8) and a subsidy given to the Merzhanov Institute of Structural Macrokinetics and Materials Science of the Russian Academy of Sciences to implement the state assignment.

**Data Availability Statement:** The data can be provided upon request.

**Conflicts of Interest:** The authors declare no conflict of interest.

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
