# Peer review of "Deflagration-to-Detonation Transition in Stochiometric Propane–Hydrogen–Air Mixtures"

_2673-3994, doi:10.3390/fuels3040040_

Round 1

Reviewer 1 Report

This paper reported on ''Deflagration-to-detonation transition in stochiometric propane–hydrogen–air mixtures''. 

Comments on the manuscript: 

***Abstract needs to be improved by adding the bacground, methodoly, results, and conclusions of this study.

***State of art needs to be better defined. 

***Introduction: This section is too long (this should be shortened to 1.5 pages). 

Author Response

We are grateful to the reviewer for valuable comments. We made our best to follow all the comments. All changes in the revised manuscript are marked in yellow.

***Abstract needs to be improved by adding the bacground, methodoly, results, and conclusions of this study.

To follow this comment, we have extended the abstract by adding several sentences with the background, methodology, results, and conclusions.

***State of art needs to be better defined. 

We have started this study several years ago and published two papers on the DDT in methane-hydrogen blends [1,2], and ethylene – hydrogen blends [3,4]. This manuscript extends the objects of investigation to propane-hydrogen blends. The actual state-of-the-art in the issue under study is highlighted in the sentence available in the original manuscript (see Introduction): “The features of DDT in such mixtures have not yet been studied in full detail.” After this sentence we refer to several relevant publications in the literature [33-37]. This is the state-of-the-art. Nevertheless, to follow this comment, we have added a sentence at the end of introduction section: “The aim of this work and the obtained results are the novel and distinctive features of the present manuscript.”

***Introduction: This section is too long (this should be shortened to 1.5 pages). 

 The Introduction section includes all relevant experimental and theoretical information on the object of our study, that is the laminar flame velocity and ignition delay of blended fuel components, as well as the blended fuel itself. Also, it includes all important studies on the developed detonations and transient combustion regimes for these components and blends. This all together provides the up-to-date background for the reader to get the idea of the manuscript. What is wrong with the length of the Introduction section if it helps understanding the subject, which is not that straightforward?

Reviewer 2 Report

I recommend publication of the manuscript once authors rectify the following error:

1. The reaction equation for propane-hydrogen-air mixture mentioned in section 2.1 is not correct. It seems that the decimals are erroneously replaced with comma signs.

Author Response

We are grateful to the reviewer for the valuable comment.

I recommend publication of the manuscript once authors rectify the following error:

  1. The reaction equation for propane-hydrogen-air mixture mentioned in section 2.1 is not correct. It seems that the decimals are erroneously replaced with comma signs.

We have replaced commas by dots, thank you.

Reviewer 3 Report

Review: fuels-1923821

The manuscript fuels-1923821 reports the "Deflagration-to-detonation transition in stochiometric 2 propane–hydrogen–air mixtures". This article shows an important topic of the engineering of chemical process, but before publishing the manuscript needs of major reviews. Therefore, we are suggesting some points below.

# The abstract of the manuscript is very simplified. I think authors must improve the abstract using the results of work.

# The text of work needs be improved regarding language. I'm suggesting to authors check the grammatical part of language (English) in article.

# The novelty of work is not clear. The novelty of article needs to be detailed in introduction of work. Some novel points can be found in the manuscript body and, thus, they can be used to describe the novelty.

# In the item 2 (Materials and Methods) of paper, at the line 146, the reaction equation needs to be numbered.

# Authors show a good number of results, but the comments of figures are not clear, and then authors need to improve.

# In the conclusion, authors need to discuss the proposed theme with its innovation proposal.

Author Response

We are grateful to the reviewer for valuable comments. We made our best to follow all the comments. All changes in the revised manuscript are marked in green.

The manuscript fuels-1923821 reports the "Deflagration-to-detonation transition in stochiometric 2 propane–hydrogen–air mixtures". This article shows an important topic of the engineering of chemical process, but before publishing the manuscript needs of major reviews. Therefore, we are suggesting some points below.

# The abstract of the manuscript is very simplified. I think authors must improve the abstract using the results of work.

We have extended the abstract by adding several sentences with the background, methodology, results, and conclusions.

# The text of work needs be improved regarding language. I'm suggesting to authors check the grammatical part of language (English) in article.

We have asked our native English-speaking colleague to check the manuscript in terms of grammar and made our best to follow the reviewer’s suggestions.

# The novelty of work is not clear. The novelty of article needs to be detailed in introduction of work. Some novel points can be found in the manuscript body and, thus, they can be used to describe the novelty.

We have started this study several years ago and published two papers on the DDT in methane-hydrogen blends [1,2], and ethylene – hydrogen blends [3,4]. This manuscript extends the objects of investigation to propane-hydrogen blends. The actual state-of-the-art in the issue under study is highlighted in the sentence available in the original manuscript (see Introduction): “The features of DDT in such mixtures have not yet been studied in full detail.” Nevertheless, to follow this comment, we have added a sentence at the end of introduction section: “The aim of this work and the obtained results are the novel and distinctive features of the present manuscript.”

# In the item 2 (Materials and Methods) of paper, at the line 146, the reaction equation needs to be numbered.

Done.

# Authors show a good number of results, but the comments of figures are not clear, and then authors need to improve.

To follow this comment, we have added some sentences to clarify our explanations.

Regarding Figure 8, we have added the arrow with text “~80ms” to the figure and noticed it in the text.

Regarding Figure 12, we have added the arrow to the figure and a sentence to the text:

“One can see that the isobars 10, 20, and 30 atm in Figure 12 are reversed at 1400 K and 0.8 (marked by the arrow), indicating that the ignition delay increases with pressure, while at lower temperatures it decreases with pressure.”

# In the conclusion, authors need to discuss the proposed theme with its innovation proposal.

To follow this comment, we have extended the Conclusions section (see revised manuscript).

Round 2

Reviewer 3 Report

Authors replied all issues this reviewer.